# Synthesis and Modeling of Ezetimibe Analogues

**DOI:** 10.3390/molecules26113107

**Published:** 2021-05-22

**Authors:** Mateo M. Salgado, Alejandro Manchado, Carlos T. Nieto, David Díez, Narciso M. Garrido

**Affiliations:** Departamento de Química Orgánica, Universidad de Salamanca, 37008 Salamanca, Spain; mateoms@usal.es (M.M.S.); alex92mc@usal.es (A.M.); eneas@usal.es (C.T.N.); ddm@usal.es (D.D.)

**Keywords:** ezetimibe, cholesterol, ligand study, pharmacophore, domino reaction, chiral amide addition, Baylis–Hillman, β-lactam

## Abstract

Ezetimibe is a well-known drug that lowers blood cholesterol levels by reducing its absorption in the small intestine when joining to Niemann-Pick C1-like protein (NPC1L1). A ligand-based study on ezetimibe analogues is reported, together with one-hit synthesis, highlighted in the study. A convenient asymmetric synthesis of (2*S*,3*S*)-*N*-α-(*R*)-methylbenzyl-3-methoxycarbonylethyl-4-methoxyphenyl β-lactam is described starting from Baylis–Hillman adducts. The route involves a domino process: allylic acetate rearrangement, stereoselective Ireland–Claisen rearrangement and asymmetric Michael addition, which provides a δ-amino acid derivative with full stereochemical control. A subsequent inversion of ester and acid functionality paves the way to the lactam core after monodebenzylation and lactam formation. It also shows interesting results when it comes to a pharmacophore study based on ezetimibe as the main ligand in lowering blood cholesterol levels, revealing which substituents on the azetidine-2-one ring are more similar to the ezetimibe skeleton and will more likely bind to NPC1L1 than ezetimibe.

## 1. Introduction

Cholesterol is an essential structural component of mammalian cell membranes and is the precursor of vitamin D, bile salts and steroid hormones; a high level of this substance in an organism can be dangerous. In fact, recent studies suggest that <70 mg/dL LDL may result in incremental cardiovascular benefits [1,2]. Additionally, raised levels of cholesterol have been associated with the onset of several diseases, such as stroke, diabetes, type 2 diabetes or endocrine disorders [3,4,5,6]. The importance of lowering cholesterol levels has been well-established. Ezetimibe is a drug that lowers plasma cholesterol levels by decreasing the absorption of cholesterol in the small intestine. Recent mouse studies have proved Niemann-Pick C1-like protein (NPC1L1) as the main intestinal cholesterol facilitator, and ezetimibe has been proved to inhibit this protein leading to a 70% reduction in intestinal cholesterol absorption [2,7,8]. When it comes to treatment, it can be used alone (marketed under the name Zetia® or Ezetrol), or combined with other medications used to lower cholesterol levels such as statins (for example, in a complex of ezetimibe/simvastatin, which is marketed as Vytorin® and Inegy). Ezetimibe is also the only adjunct to statin therapy that has successfully shown cardiovascular benefits when combined. This association of ezetimibe makes it possible to reduce the doses of statin. Taking into account all of these features, an intensive synthetic study is here devoted to this drug and its derivatives.

## 2. Results

Below, we describe the optimized synthesis of β-lactam 1, one of the most promising candidates when it comes to ezetimibe analogues (see Docking section of this article). All the compounds analyzed in this section are accessible by means of the methodology developed by the group, which is underlined below.

We have demonstrated the use of chiral lithium amide (α-methylbenzyl)benzylamide (*R*)- or (*S*)-2 in different domino reactions [9,10,11,12]. Firstly, in 1997, we published an asymmetric conjugate addition cyclization of octa-2,6-diendioate I, initiated by a chiral lithium amide, to obtain, stereoselectively, the methyl 2-amino-5-(2-methoxy-2-oxoethyl)cyclopentane-1-carboxylate (II Scheme 1), [13,14,15] and applied it to the synthesis of (*R*) and (*S*)-methyl (2-methoxycarbonylcyclopent-2-enyl)acetate (III and IV) and (*R*)- and (*S*)-2-(2-hydroxymethylcyclopent-2-enyl)ethanol, useful homochiral synthons for monoterpenes [13], and to the asymmetric synthesis of all the stereoisomers of 2-amino-5-carboxymethyl-cyclopentane-1-carboxylic acid (V Scheme 1) [14,15]. Most recently, we performed a multicomponent domino reaction, yielding VI, which has been applied towards the asymmetric synthesis of cyclopentane[c]pyran core of iridoid natural products (VII Scheme 1) [16]. 

We later showed, in 2008, a novel domino reaction (allylic acetate rearrangement, stereoselective Ireland–Claisen rearrangement and asymmetric Michael addition) [17]; a protocol starting from Baylis–Hillman adducts VIII and IX, using chiral lithium amide (*R*)-2, afforded *δ*-amino acids X, which are able to be transformed into piperidines XIV throughout piperidone XIII. Interestingly, in this reaction, the derivative IX provides better yield of X than Baylis–Hillman adduct VIII [17,18]. Convenient substitution in piperidone XIII allows the synthesis of piperidine dicarboxylic acid (PDA), [18] as well as neurokinin analogues, (+)-L-733,060 (+)-CP-99,994 [19]. It must be noted that using this methodology, PDA compounds were able to be obtained, starting from cinnamaldehyde and methyl crotonate, to obtain the Baylis–Hillman adduct [18], in which the cinnamaldehyde double bond is a masked carboxylic functionality. Recently, this methodology was applied to the synthesis of 2,3,6-trisubstituted piperidines XII (Scheme 2) by converting the acid X to an aryl ketone XI prior to debenzylation-cyclization one-pot reactions [20].

Herein, following the aforementioned methodology, we report the synthesis of azetidine-2-one derivative, as shown in Scheme 3, an analogue of ezetimibe, starting from *p*-anisaldehyde and tert-butyl acrylate, which are used in the Baylis–Hillman reaction. In this manner, the adduct 3 was obtained, an α,β-unsaturated ester previously synthetized by Brand et al. [21] in acceptable yield using phenol as an additive to improve reaction kinetics [22,23]. This alcohol is acetylated with acetic anhydride and pyridine, giving the acetate 4, which was treated with the chiral amide (*R)*-2 to give 5 with good yield and excellent diastereomeric excess, higher than 95%. The strategy for obtaining the β-amino acid from the *δ*-amino acid is based on simple transformations of the different carboxylates of 5. Treatment of 5 with TMSCHN_2_ gives rise to 6 in quantitative yield, and its structure was corroborated by the appearance of a signal at 3.68 ppm (s, 3H, COOMe). The subsequent release of the tert-butyl ester in an acid medium leads to the β-amino acid 7 with an excellent yield. Compound 7 is treated with ceric ammonium nitrate (CAN), which produces a chemoselective debenzylation of the least substituted group, obtaining 8 with 84% yield. Once the β-amino acid has been obtained, the lactam coupling is carried out in order to obtain the final compound 1. In this way, a combination of DIPEA, 1-hydroxybenzotriazole hydrate (BtOH) and EDCI is used under room temperature. In this case, 1-hydroxybenzotriazole is used as a carboxylic acid activating group for the entry of carbodiimide, since it reacts rapidly with *o*-acylisourea, also avoiding side reactions and risks of racemization [24,25,26]. In addition, the formed urea can be removed by successive washings. This fact makes it easy to use this methodology in solid phase synthesis.

Although all NMR signals as well as reaction details are widely explained in the supporting information section, see Appendix A, the main ^1^H NMR signals of the ezetimibe analogue 1 are: 4.88 ppm (q, *J* = 7.2 Hz, 1H, C(α)H), 1.33 ppm (d, *J* = 7.2 Hz, 3H, C(α)Me) and 7.34–7.19 ppm (m, 5H, ArH) due to the α-methylbenzyl group; 3.70 (s, 3H, -COOMe) and 3.47 (s, 3H, -OMe), corresponding to the methoxy groups attached to ester and phenyl functions, respectively; 7.02 (d, *J* = 8.7 Hz, 2H) and 6.74 (d, *J* = 8.7 Hz, 2H), showing an aromatic AB system, and 4.43 (d, *J* = 5.6 Hz, 1H, H4) and 3.22 (td, *J* = 8.3, 5.6 Hz, 1H, H3), due to the *cis*-disubstituted β-lactam core, ratifying its synthesis.

### 2.1. Pharmacophore Docking Overlay 

Although several techniques can be carried out in order to obtain in silico results when it comes to the computational modelling of ligands (a ligand is a defined polyatomic molecular entity with the experimental or predicted capability to bind a central macromolecular entity called the target) [27], such as in studies based on the target, which focus on the protein receptor, this study is based on the pharmacophores, focusing on the comparison of the pharmacophore groups, representing the spatial arrangement of characteristics that are essential for a molecule that interacts with a specific target receptor, of a known ligand, in this case ezetimibe, with the compounds subjected to this study.

### 2.2. Design of the Ligands

A bibliographic search of the main ezetimibe pharmacophores [28,29,30,31,32,33,34,35] was carried out, and it was found that the following requirements are advisable to obtain an analogue with reasonable activity:

An azetidinone ring is required in the structure, Figure 1.

In position 1 of the ring, it is advisable to have an *N*-phenyl or *N*-benzyl group, although a range of other substituents are allowed.An oxygenated function is required at a distance of three carbons from the cycle at position 3.An aromatic group is required at position 4, preferably *p*-hydroxyl or *p*-methoxyphenyl. In addition, isomers with *S* stereochemistry at this carbon show greater activity against *R*.The 3*S* and 3*R* isomers have comparable activity without preference. That is, there is no stereochemical preference at this asymmetric center [30]. This point is relevant in the present work and it has been very well documented by J.W. Clader et al., in the article with tittle: “2-Azetidinone Cholesterol Absorption Inhibitors: Structure-Activity Relationships on the Heterocyclic Nucleus” [35].

With these guidelines, we decided to study derivatives with *cis* stereochemistry of the *β*-lactam ring, since this stereochemistry is accessed directly in our synthetic pathway, as shown above. However, this work could be expanded in the future by examining the ligands with the substituents of the ring in *trans*, easily accessible by means of a similar methodology. The oxygenated substituents at position 3 are also modified. Additionally, the substituent R^1^ will be modified in order to find the best groups, as well as the aromatic function at four position. This way, in Table 1, ligands L1–L10 are presented.

Once the ligands were designed, the study was carried out by the PharmaGist web server [36,37,38], which detects the pharmacophore groups of a set of molecules, see Table 2. The method used is based on ligands, and therefore it does not require the structure of the target receptor. Instead, the input files are a set of drug-like molecule structures that are known to bind to the receptor. We calculated the pharmacophores by multiple flexible alignments of the input ligands and assigned a score according to the goodness of the spatial characteristics and the overlap of structures. The main innovation of this approach is the flexibility of the input ligands, which is handled explicitly and deterministically in the alignment process. Another important feature of the method is the ability to detect pharmacophores shared by different subsets of input molecules. This ability is a key advantage when the ligands belong to different binding modes or when the input contains outliers in the score.

Once scores (score is a theoretical value, obtained from a resulting pairwise alignment, and is a weighted sum of the matched pivot features. These pivot features are the detected pharmacophores) are presented, L1 is highlighted, whose structure includes an aromatic interaction, possesses a hydrogen bond donor group and also hydrogen bond acceptor groups.

In Figure 2, we can see the overlaps of ligand L1 with ezetimibe (Scheme 1). It can be seen how the aromatic rings are superimposed on everything that contains the hydroxyl of phenol. We can also see how the carboxymethyl group overlaps with the hydroxyl of the ezetimibe chain, and both groups have electronegative atoms (Oxygen). Furthermore, the β-lactam rings and the other hydrogen acceptor groups overlap, despite having different relative stereochemistry in the β-lactam ring.

It would be interesting to carry out biological tests of this class of ligands, which would allow a more precise understanding of the effects that have been discussed and could also help the homological construction to obtain higher quality data in the subsequent studies that are carried out.

## 3. Conclusions

A useful method for the enantioselective synthesis of compounds with interesting hypolipémiant properties is described. By using the *p*-methoxy-benzaldehyde Baylis–Hillman adduct in an efficient domino reaction (allylic acetate rearrangement, stereoselective Ireland–Claisen rearrangement and asymmetric Michael addition), we have ready access to *δ*-amino acids 5 with 59% yield. A subsequent reversal of ester and acid functionality paves the way to the lactam core after monodebenzylation and lactam formation under activation for the attainment of amide in 70% yield and four steps. It is important to note that the series of analogous reactions using the enantiomer of lithium amide (*S*)-2 in the domino reaction will allow easy access to the enantiomer of the above-mentioned compounds, as has been demonstrated in previous work with the synthesis of (*R*)- and (*S*)-cyclopentancarboxylates III and IV, using the enantiomerics amides 2. Taking this into account and considering the ease of deprotection and new functionalization of *N* in position 1 of azetidin-2-one core, the possibility of using different starting aldehydes in the Baylis–Hillman reaction that account for C-4 substitution and to take advantage of the reactivity of the ester moiety within the C-3 chain, all this will provide diversity to the methodology for quick access to libraries of molecules with an emphasis on SAR studies of this series of analogues based on the support for docking studies. Further work is being undertaken in our laboratory.

## Data Availability

Data is contained within the article and supporting material.

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
