# Peer review of "Synthesis and Modeling of Ezetimibe Analogues"

_molecules, 2021, doi:10.3390/molecules26113107_

Round 1

Reviewer 1 Report

This paper described the synthesis of (2S,3S)-N-(alpha)-(R)-methylbenzyl-3-methoxycarbonylethyl-4-methoxyphenyl beta-lactam from p-methoxy-benzaldehyde through 8 steps. Intermediates 2-7 and final product 8 are novel compounds, which are characterized by 1H NMR, 13C NMR, IR and HRMS. The overlaps of ligand L1 ((2S,3S)-N-(alpha)-(R)-methylbenzyl-3-methoxycarbonylethyl-4-hydroxyphenyl beta-lactam) with Ezetimibe was conducted based on its pharmacophore, which indicated that L1 can have an aromatic interaction, possess a hydrogen bond donor group and also hydrogen bond acceptor groups. However, there are some mistakes or defects in this paper, which are listed as follows.

  • The intermediate 2 is a known compound, which was published in the reference (Brand, Maja; Drewes, Siegfried E.; Loizou, Georgia; Roos, Gregory H. P. Asymmetric synthesis of 2-​substituted acrylate esters. Synthetic Communications 1987, 17(7), 795-802).
  • The final compounds L1-L10 should be confirmed by 1H NMR, 13C NMR, IR and HRMS besides L1.
  • L1-L10 should be tested their biological activities, and then structure-activity relationship can be discussed in the context of this paper.
  • R1 group of Table 1 in page 5 should be R2, and R2 group of Table 1 in page 5 should be R1.
  • In the supporting information, 1H RMN and 13C RMN should be 1H NMR and 13C NMR, respectively. In addition, there are some errors in English grammar. Check them!

In a word, after above mistakes are corrected, this paper can be acceptable by Molecules.

Author Response

This paper described the synthesis of (2S,3S)-N-(alpha)-(R)-methylbenzyl-3-methoxycarbonylethyl-4-methoxyphenyl beta-lactam from p-methoxy-benzaldehyde through 8 steps. Intermediates 2-7 and final product 8 are novel compounds, which are characterized by 1H NMR, 13C NMR, IR and HRMS. The overlaps of ligand L1 ((2S,3S)-N-(alpha)-(R)-methylbenzyl-3-methoxycarbonylethyl-4-hydroxyphenyl beta-lactam) with Ezetimibe was conducted based on its pharmacophore, which indicated that L1 can have an aromatic interaction, possess a hydrogen bond donor group and also hydrogen bond acceptor groups. However, there are some mistakes or defects in this paper, which are listed as follows.

We appreciate your thoughtful review of our paper. Each of your comments is appropiate and useful for improving the paper. Below we respond to echa comment and indicate how we plan to revise the manuscritp accordingly. For clarity, we have copied your comments in their entirety in normal font and followed each with our response in blue colour directly below.

  • Comment 1. The intermediate 2 is a known compound, which was published in the reference (Brand, Maja; Drewes, Siegfried E.; Loizou, Georgia; Roos, Gregory H. P. Asymmetric synthesis of 2-​substituted acrylate esters. Synthetic Communications 1987, 17(7), 795-802).

Response 1. Corrected.

The article has been introduced as a new reference 21 and cited in the text.

  • Comment 2. The final compounds L1-L10 should be confirmed by 1H NMR, 13C NMR, IR and HRMS besides L1.
  • Comment 3. L1-L10 should be tested their biological activities, and then structure-activity relationship can be discussed in the context of this paper.

Response 2 and 3. We have currently synthesized compound 1 and demonstrated the direct synthetic pathway for ligands L1-L10, which have been evaluated by docking. In addition, the methodology allows the synthesis of different derivatives with different stereochemistry, among which ezetimibe is included, and once the work is completed it will be published in due course.

  • Comment 4. R1 group of Table 1 in page 5 should be R2, and R2 group of Table 1 in page 5 should be R1.

Response 4. Corrected.

The referee is correct and this was an error for which we apologise.

  • Comment 5. In the supporting information, 1H RMN and 13C RMN should be 1H NMR and 13C NMR, respectively. In addition, there are some errors in English grammar. Check them!

Response 5. Corrected.

We appreciate the careful review.

We have carefully checked the English grammar and have done our best,  including proofreading by a native speaker, hopefully the current manuscript meets the standards. 

Reviewer 2 Report

Please consider the attached file

Author Response

A convenient total asymmetric synthesis of an azetidinone derivative, extensible also to similar compounds, is illustrated, starting from a precursor obtainable thanks to a domino process already applied by the authors to build up other scaffolds. These new derivatives are analogues of Ezetimibe, an important anticholesterolemic drug, so an in silico study was prompted to evaluate the similarity degree between new compounds and the lead. 

The synthetic aspect of the manuscript in my opinion is surely of value, so it deserves publication, but many corrections and changes are necessary to make the manuscript more complete and clear, so to be suitable for Molecules standard. 

We appreciate your thoughtful review of our paper. Each of your comments is appropiate and useful for improving the paper. Below we respond to each comment and indicate how we plan to revise the manuscritp accordingly. For clarity, we have copied your comments in their entirety in normal font and followed each with our response in blue colour directly below.

  • Comment 1. First of all, the manuscript is written in an uncorrect English language, with normally unused words, and some sentences result hardly comprehensible. A thorough revision is needed in this sense.

Response 1. We have carefully checked the English grammar and have done our best,  including proofreading by a native speaker, hopefully the current manuscript meets the standards. 

  • Comment 2. Lines 51-60 describe with too many details a previous study (1997), furthermore with no graphical help, that should be just cited with bibl ref. Instead, just a mention was made to the domino reaction discovered in 2008 and employed before to obtain piperidine derivatives, and here to get the precursor of the target. I think that some more explanation would be useful, for “classic reactions” that are not so common. In scheme 1, the caption says “retrosynthesis”, but the arrows used in the Scheme are not the correct ones. 

According to your kind suggestion,  we have introduced a new scheme 1, to clarify the reactions mentioned and demonstrating that different enantiomeric amides lead to enantiomeric products and even implement the methodology that allows obtaining the 8  diastereoisomers (V, scheme 1), taking advantage of the complementarity of the enantiomeric amides in successive additions.

It also includes greater detail of the domino reaction with the Baylis-Hillman adducts and we have used the appropriate arrows to indicate the retrosynthesis.

  • Comment 3. Furthermore, it would be advisable to use superscripts, rather than subscripts, to differentiate “R”.

According to your kind suggestion,  this have been corrected.

  • Comment 4. The following lines describe the synthetic procedure; full names should be mentioned for all the reagents.

According to your kind suggestion,  this have been corrected.

  • Comment 5. 

    The most problematic part is the “docking” paragraph, that appears rather obscure under many aspects. I think that a reader not directly involved in this kind of study may find it difficult to understand the significance of the reported data.

    Q: When the authors say “ligands”, what do they exactly mean?

    A: In the context of Molecular Modelling, a ligand is a defined polyatomic molecular entity with the experimental or predicted capability to bind a central macromolecular entity called target. [1]

    Q: Would not be better to say “substituents”?

    A: Substituents are referred to polyatomic sub-molecular entities, which represent sections of a molecule.

    Q: In table 2, what is intended for “score”?

    A: Score is a theoretical value, obtained from a resulting pairwise alignment, is a weighted sum of the matched pivot features. These pivot features are the detected pharmacophores.

    Q: What would be the maximum value? can nine be considered a good score? Maybe it could be better to furnish a %, or a fractional value (9/10 or rather 9/100 or 1000?).

    A: Score is a relative parameter depending on the defined reference input (Ezetimibe this case), not universal one. Changing the reference molecule will obviously change the Score result. In this regards, the higher the value the closer similarity to the reference. In other hand, it is essential to maintain the consistency of Software output parameters across literature, so it is recommended to keep the output parameters as they are avoiding confusion upon modifying them.

    Q: Some explanation about the data in the columns should be furnished; in particular what does represent + and – (all zeros) reported in the last two columns?

    A: Table 1 description has been updated clarifying all the table fields. [2]

    Table 1. Results of the superposition with Ezetimibe A. CE: Spatial Characteristics, Ar: Aromatic (aromatic rings), Hy: Hydrophobic (prominent hydrophobic groups), Do: Donors (Hydrogen Bond Donors), Ac: Acceptors (Hydrogen Bond Aceptors), +: Positive (cationic centers), -: Negative (anionic centers).

    [1]. IUPAC. Compendium of Chemical Terminology, 2nd ed. (the "Gold Book"). Compiled by A. D. McNaught and A. Wilkinson. Blackwell Scientific Publications, Oxford (1997). Online version (2019-) created by S. J. Chalk. ISBN 0-9678550-9-8. https://doi.org/10.1351/goldbook.

    [2] Nucleic Acids Research, Volume 36, Issue suppl_2, 1 July 2008, Pages W223–W228,

According to the referee's kind suggestions, some discussion is introduced into the text and references shown above.

  • Comment 6. In the conclusions paragraph, the English becomes still worse than usual. The assumption that the enantiomer of the product could be obtained just changing the starting lithium amide configuration should be demonstrated. 

Response 6. We agree with the referee and apologise for that.

We appreciate the careful review and most of the conclusion have been rewritten trying to do our best.

We have already demonstrated that enantiomer products are obtained by changing the lithium amide configuration as indicated in scheme 1 and reference 13. Even when this has not been carried out in the present work, this fact has been revealed in different works that we have published. If the referee considers it acceptable we would like to keep it as a demonstration of the synthetic diversity that it provides. 

  • Comment 7. Corrections are needed also in the supplementary material.
  • First step. Does the reaction really require 29 days? Some confusion was done about the reported value for the grams obtained of compound 2. 

A: It is certainly a slow kinetic reaction, that is why it is necessary to use phenol as a catalyst for the reaction, this time it was allowed 29 days, because the excess time did not reduce the yield and coincided with other jobs to be completed and a period of absence in the laboratory.

Value of grams for compound 2, have been checked and corrected.

  • In the 1H NMR description of 5, probably the signal 2.28-2.11 is not to be counted. In the molecular formula, oxygen atoms are 5 and not 3.

The referee is correct, the signal is deleted and the molecular formula have been corrected.

  • The names proposed for 6 and for 7 are not acceptable as far as 1-hydrogen…. is concerned . The number of protons in 1H NMR of 6 does not seem correct. In that of 7, the first m should count 5 H, not 12! 

We have checked this and given the name indicated by chemdraw.

Right it is 5 H and have been corrected

  • In general, for some 13C, the number of signals does not correspond to that expected. 

All NMR 13C have been checked and corrected accordingly when necesary.

We appologized that compound 7 contains small amount of solvent (EtAcO).

Reviewer 3 Report

This manuscript describes a piece of works consistent with Garrido’s group continuing studies.  As presented,  the docking study is a novel development.  Ezetimibe is a cholesterol transport inhibitor, although the usage is limited, compared with the widely applied statin drugs as the first selection.  The present work deals with chiral synthesis of an Ezetimibe analogue and its docking study.    

The synthesis utilizes Baylis-Hillman, Ireland-Claisen rearrangement, and asymmetric Michael sequences, which were previously reported by the author’s group.  The docking study was implemented by the open PharmaGift web site.  Compared with the effective in vivo and/or in vitro experiments, the obtained outcomes are somewhat incredible.       

On the whole, the reviewer recommends the publication in Molecules after major revisions commented below.

  • Results section describes too many self-citing papers (Refs. 9-20). The author should concentrate more critical self-works, and moreover, brief and general viewpoints in this area are required in an additional scheme.
  • The word of “total synthesis” (Abstract) is solely applied for the case of natural products. The present analogue does not correspond to this case.
  • In Scheme 1, the target new compound “8” did not appear in the text sentence. In addition, the compound numbering order should appear on first come, i.e., “1
  • In Scheme 1, “≡” means congruence. This expression should be altered to another type, otherwise, deleted.
  • Superscripts, italics, etc. should be carefully checked in the whole sentences and schemes.
  • all NMR signals → NMR monitoring study
  • Figure 1: “oxygen moiety should be placed in the formula.
  • In “Design of the ligands,” the reason for 3S and 3R isomers have comparable (described in Refs. 26-32) is commented. Chiral discrimination is the major topic in SAR.  The author should explain brief statement of this issue in the text.  Although the α-position in carbonyl compounds have considerable racemization prone, the relevant β-lactam antibiotics have rigorous stereogenic centers.          

Author Response

This manuscript describes a piece of works consistent with Garrido’s group continuing studies.  As presented,  the docking study is a novel development.  Ezetimibe is a cholesterol transport inhibitor, although the usage is limited, compared with the widely applied statin drugs as the first selection.  The present work deals with chiral synthesis of an Ezetimibe analogue and its docking study.    

The synthesis utilizes Baylis-Hillman, Ireland-Claisen rearrangement, and asymmetric Michael sequences, which were previously reported by the author’s group.  The docking study was implemented by the open PharmaGift web site.  Compared with the effective in vivo and/or in vitro experiments, the obtained outcomes are somewhat incredible.       

On the whole, the reviewer recommends the publication in Molecules after major revisions commented below.

We appreciate your thoughtful review of our paper. Each of your comments is appropiate and useful for imporoving our paper. Below we respond to each comment and indicate how we plan to revise the manuscript accordingly. For clarity, we have copied your comments in their entirety in normal font and followed each with our response in blue colour directly below.

  • Q: Results section describes too many self-citing papers (Refs. 9-20). The author should concentrate more critical self-works, and moreover, brief and general viewpoints in this area are required in an additional scheme.

A: We agree with the referee, our intention was to outline the achievement of the new procedures that are used in the article to get the objectives, in this sense a new scheme has been included for clarity. 

  • Q: The word of “total synthesis” (Abstract) is solely applied for the case of natural products. The present analogue does not correspond to this case.

A: According to your kind suggestion, total have been omitted throughout the text.

  • Q: In Scheme 1, the target new compound “8” did not appear in the text sentence. In addition, the compound numbering order should appear on first come, i.e., “1

A: We agree with the referee and the compound appears as 1 in scheme 1 and is cited in the text .

  • Q: In Scheme 1, “≡” means congruence. This expression should be altered to another type, otherwise, deleted.

A: This sign has been deleted and ezetimibe underline to show similarity with 1.

  • Q: Superscripts, italics, etc. should be carefully checked in the whole sentences and schemes.

A: This aspect have been carefully checked and corrected accordingly.

  • Q: all NMR signals → NMR monitoring study

A: Corrected.

  • Q: Figure 1: “oxygen moiety should be placed in the formula.

A: Based on your kind suggestion, oxygen is in the formula (fig. 1) and the substituents in the table1 have been modified accordingly. 

  • Q: In “Design of the ligands,” the reason for 3S and 3R isomers have comparable (described in Refs. 26-32) is commented. Chiral discrimination is the major topic in SAR.  The author should explain brief statement of this issue in the text.  Although the α-position in carbonyl compounds have considerable racemization prone, the relevant β-lactam antibiotics have rigorous stereogenic centers.  

A: We agree totaly in relation to stereogenic centers in β-lactam antibiotics and biological activity. But in relation to ezetimibe analogues, a SAR rigurous study  on the hererocyclic nucleus has been done by J. W. Clader, with very interesting result that extraordinarily validates our methodology, but above all the conclusion that C-3 configuration is not relevant for activity as we have direct access to the cis derivative even when the trans could be readily prepared. This reference and the discussion have been included in the manuscript.

Round 2

Reviewer 1 Report

1H NMR and 13C NMR of compounds L2-L10 should be provided in the supporting information. After above additon, this paper can be acceptable.

Author Response

We appreciate your thoughtful review of our paper, It has definitely helped us a lot to correct some errors and concepts throughout it.

All changes in the corrected version of the manuscript, marked with a yellow background.

English language  and  style have been corrected with the help of a native English-speaking colleague. In this regard, we regret that despite the effort initially made, our level of English was not able to correct some deficiencies in this regard, and  in future ocassions if necessary we will use mdpi editing service.

In relation to 1H and 13C NMR of compounds in table 1,

The idea of the article within this special issue "New Synthetic Methods for Organic Compounds" is to show how by a novel domino reaction (new synthetic method) beta-lactams similar to ezetimibe can be synthesized. This has been achieved and this way we demonstrate how to get different derivatives. At present it is not possible to synthesize all the compounds in Table 1 since the doctoral student who carried out the study is doing post-Doc at another University and current studies are focused on the synthesis of ezetimibe using this methodology. Once the utility of the methodology will be reported, so that, the scientific community can take advantage of it. 

Reviewer 2 Report

Sir,

the second version of the manuscript

 Synthesis and modeling of Ezetimibe analogues

By

Mateo M. Salgado1, Alejandro Manchado1, Carlos T. Nieto1, David Díez1, Narciso M. Garrido1*

keeping into account my comments, and ameliorated in English form and clarity, is in my opinion certainly worth of publication. I just have to recommend some further little corrections:

line 51: after the compound name, please indicate I

lines 53 and 55: please correct the name of II as methyl 2-amino-5-(2-methoxy-2-oxoethyl)cyclopentane-1-carboxylate and that of III and IV as methyl (2-methoxycarbonylcyclopent-2-enyl)acetate; in the following name, please remove hyphen between methyl and cyclopent

line 105: details are widely explained …. Information section, the main …

line 180: SC rather than CE, I suppose, abbreviates spatial characteristics; otherwise in the column, change SC with CE. Just out of curiosity, I would like to know if the obtained score is good: which could be a particularly satisfying score?

Line 201 and following: now conclusions are more detailed but some periods are in my opinion too long and complicated; for instance, lines 204-205, the three step could be put into round brackets (allylic …… Michael addition); line 212: no space between cyclopentan and carboxylates; line 213: “considering” could sound better than “dealing with”; the easy deprotection and …; the possibility to use different starting aldehyde (that account ….) and to take advantage of ..

I think “Further work is undertaken in our laboratory” could be the conclusion; the following seems unnecessary.

Supporting:

compound 8, change the character of the name of compound; the first signal in the 1H NMR counts 5 H (just Ph hydrogens!)

paragraph 7. Please remove the final “e” of “betalactame”; also for this compound, the first signal in the 1H NMR counts 5 H (just Ph hydrogens!)
